# Rapid synthesis of phosphor-glass composites in seconds based on particle self-stabilization

Yongsheng Sun[1], Yuzhen Wang[1], Weibin Chen[2], Qingquan Jiang[2], Dongdan Chen[2], Guoping Dong [2] & Zhiguo Xia [1] ✉

Phosphor-glass composites (PGC) are excellent candidates for highly efficient and stable photonic converters; however, their synthesis generally requires harsh procedures and long time, resulting in additional performance loss and energy consumption. Here we develop a rapid synthetic route to PGC within about 10 seconds, which enables uniform dispersion of $Y_3Al_5O_{12}:Ce^{3+}$ (YAG:Ce) phosphor particles through a particle self-stabilization model in molten tellurite glass. Thanks for good wettability between YAG:Ce micro-particles and tellurite glass melt, it creates an energy barrier of $6.94 \times 10^5$ zJ to prevent atomic-scale contact and sintering of particles in the melt. This in turn allows the generation of YAG:Ce-based PGC as attractive emitters with high quantum efficiency (98.4%) and absorption coefficient (86.8%) that can produce bright white light with luminous flux of 1227 lm and luminous efficiency of 276 lm W$^{-1}$ under blue laser driving. This work shows a generalizable synthetic strategy for the development of functional glass composites.

Rare earth phosphor materials as photonic converters, exampled by commercial $Y_3Al_5O_{12}:Ce^{3+}$(YAG:Ce), $CaAlSiN_3:Eu^{2+}$, and $Sr[LiAl_3N_4]:Eu^{2+}$, etc., have received broad attentions in lighting and display[1–4], sensor[5,6], and photovoltaic cells[7,8]. Conventional color converters in optoelectronic devices are fabricated by luminescence materials embedded in transparent organic polymers, which have been favored due to their convenience and universality, but the organic encapsulation strategy faces major obstacles of irreversible degradation under heat, humidity, and strong blue/ultraviolet light radiation[9]. Especially with the rise of high-power laser-driven (LD) devices, phosphor-polymer composites cannot meet the actual demands. Rare earth-doped single-phase transparent ceramics and crystals are considered as effective all-inorganic color converters that can overcome the limitations of organic encapsulation[10–12]. However, the high cost and limited bulk materials make it difficult to achieve large-scale production and application.

Synthesis of phosphor-glass composites (PGC) are regarded as a more convenient and economical strategy for photonic converters due to the abundant selectivity of the phosphor and matrices materials[13], and improved heat dissipating capability[14]. Nevertheless, the dominant fundamental issues for the development of photonic conversion composites, as in other composite materials, are maintaining the intactness of the embedded phosphors and preserving luminescent properties[15–17]. Typically, three routine trains of thought are devoted to addressing these synthetic problems, including reducing the synthetic temperature, selecting a suitable matrix material, and adopting a fast synthesis strategy. Thereinto, the first two routes are the focuses of current research. For example, some reports utilize low-sintering temperature ceramics ($CaF_2$[14] and hydroxyapatite[18]) and low-melting temperature glass (tellurite[19,20] and phosphate glasses[21,22]) as matrices to reduce the synthesis temperature and rely on hot isostatic pressing[23], gas pressure sintering[24], and spark plasma sintering technology[25] to suppress interface reactions. Moreover, Qiu et al.

[1]State Key Laboratory of Luminescent Materials and Devices, Guangdong Provincial Key Laboratory of Fiber Laser Materials and Applied Techniques, Guangdong Engineering Technology Research and Development Centre of Special Optical Fiber Materials and Devices, School of Physics and Optoelectronics, South China University of Technology, Guangzhou 510641, China. [2]School of Materials Science and Engineering, South China University of Technology, Guangzhou 510641, China. ✉e-mail: xiazg@scut.edu.cn

discussed the intrinsic inhibition of interfacial reactions in PGC using silica glass instead of multi-component glass as the matrix[13,26]. Across all reported phosphor-glass/ceramic composites, a conspicuous disadvantage in regard to their synthetic procedure is the considerable time requirements necessary to form bulk composites. And the long sintering time generally gives rise to an unavoidable thermal erosion and performance degradation of composites. Accordingly, the fast synthesis strategy that can fabricate the PGC in seconds is urgently needed for ensuring the completeness of the embedded particles but also greatly reduces the cost and improves the efficiency of manufacture[27].

Here, we show a synthetic protocol in seconds based on particle self-stabilization model in glass melt, which good wettability between the molten glass and particles creates an energy barrier of $6.94 \times 10^5$ zJ, preventing atomic-scale contact and sintering of particles in the melt, to achieve a dense uniform dispersion of YAG:Ce phosphor particles in tellurite glass within about 10 s. The YAG:Ce based PGC not only possesses high quantum efficiency (98.4%) and absorption coefficient (86.8%), but also produces desirable white light with luminous flux of 1227 lm and luminous efficiency of 276 lm W$^{-1}$ under blue laser excitation. More importantly, the fast synthetic strategy could also be extended to synthesize various color-tunable composites with high quantum efficiency (>94.0%).

## Results

### Synthesis and microstructure characterization of YAG:Ce-PGC

Glass generally exhibits stable physical/chemical properties and good transparency in the ultraviolet to near infrared region, which is used as excellent photo-functional composite matrix[28–30]. However, conventional synthesis and processing methods of PGC suffer from serious thermal erosion, low synthesis rate, and tedious process. The glass with low melting point and low melt viscosity acting as composite matrix is a good choice to solve these issues. The melting temperature of glass is related to the polarizability of its constituent cations. The higher the cation polarizability, the lower the melting temperature of the glass[31]. Cations in the outer layer that contain non-inert electron pairs (such as $Pb^{2+}$, $Bi^{3+}$, $Te^{3+}$, etc.) and an 18-electron configuration (such as $Zn^{2+}$) have higher polarizability[32]. Here, tellurite glass with the composition of 75 $TeO_2$–15 $ZnO$–10 $Na_2O$ is screened and used, which has a low viscosity (0.1564–0.0373 pa·s) at relatively low temperatures (600–800 °C) offering the endless possibility for all-inorganic functional composites (Supplementary Fig. 1).

As shown in Fig. 1a, the synthesis of PGC includes two processes of rapid dispersion and fast cooling, and the preparation is completed in about 10 s, which are attractive compared to previous reports on phosphor glass/ceramic composites[13,33,34]. The experimental procedure was simple, as described below: we firstly poured commercial YAG:Ce phosphor particles with different contents (2–20 wt%) into tellurite glass melt at 650 °C, and then we used quartz glass rods for simple agitation to disperse them uniformly (Supplementary Movie 1). And finally the mixtures are quenched quickly to obtain block YAG:Ce-PGC (Supplementary Fig. 2). The obtained YAG:Ce-PGC sample has good macroscopic uniformity, an overall appearance of bright yellow under indoor light, and intense emission under 450 nm excitation (Fig. 1b). Under the optical microscope, YAG:Ce phosphor particles are densely and uniformly dispersed in the tellurite glass matrix without agglomeration (Fig. 1c and Supplementary Movie 2). With such a fast preparation process, this dispersion result is incredible. And the refractive index difference ($\Delta n$) between tellurite glass (1.97) and YAG:Ce crystal (1.84) is 0.13 (Supplementary Table 1). Thus, the uniform dispersion of YAG:Ce particles, the small $\Delta n$, and few pores create a decent transmittance (about 40%) of YAG:Ce-PGC sample (Fig. 1d and Supplementary Fig. 3), even if the doping content reaches 10 wt%. In addition, the moderate light scattering is beneficial for the color uniformity and the light extraction of high-power light-emitting diodes

(LEDs), especially when LDs are used as the excitation source[35]. Considering the YAG:Ce particle size distribution before and after embedding in tellurite glass (Fig. 1e and Supplementary Fig. 4), the size of YAG:Ce particles remains constant (about 10 μm). The 3D reconstructed confocal laser scanning microscope (CLSM) image (Fig. 1f) also proves that YAG:Ce particles are equably dispersed in tellurite glass. More importantly, another advantage of this strategy over those pressure-assisted methods is that it can be formed into various shapes, even fibers, by different quenching and fabrication processes, to facilitate the application of various scenarios in the future (Supplementary Fig. 5).

To investigate the interfacial stability between YAG:Ce particles and tellurite glass, we firstly measured the X-ray diffraction (XRD) patterns of YAG:Ce-PGC. Obviously, all diffraction peaks are coincident to YAG phase suggesting that the incorporation of YAG:Ce did not induce the precipitation of other crystalline phases (Supplementary Fig. 6). Subsequently, the local structures of YAG:Ce-PGC were evaluated by Raman vibrational modes (Supplementary Fig. 7). The Raman spectra of the YAG:Ce-PGC fabricated at 650 °C, with a superposition of those from tellurite glass and YAG:Ce, represents a pivotal signal that the interface reaction is absent during rapid synthesis. The interfacial stability of YAG:Ce particles dispersed in tellurite glass at nanometer scale was further characterized by high-resolution transmission electron microscopy (HRTEM), selected area electron diffraction (SAED), and energy dispersive spectroscopy (EDS) mapping analysis. As illustrated, a boundary line between crystal and amorphous phases is explicit, and HRTEM images and SAED patterns confirm the cubic garnet phase of YAG:Ce and the amorphous phase of tellurite glass (Supplementary Fig. 8). From the high-angle annular dark-field scanning transmission electron microscopy (HAADF-STEM) image, a well-defined boundary can be also observed, which is manifested by the EDS mapping profile, showing significant elemental differences and no observable element diffusion region between the YAG:Ce crystal and tellurite glass (Fig. 1g). These results indicate that interfacial interactions between YAG:Ce particles and tellurite glass are negligible, which is in stark contrast to the thickness of about 50–300 nm in the previous studies[13,26,36,37], and it further proves that our synthesis method is a nondestructive composite strategy.

### Verification of particle self-stabilization model

To investigate the dispersion stability of YAG:Ce particles in tellurite glass melt, we put the glass melt with uniformly dispersed YAG:Ce particles back into the high-temperature furnace to continue heating at 650 °C and took it out at different holding times, quenched, and formed. The YAG:Ce particles were uniformly dispersed even after our samples remained in the liquid state for about 120 s without agitation processing (Supplementary Fig. 9). This provides convincing evidence that YAG:Ce particles realize good dispersion and self-stabilization in molten tellurite glass before solidification. To better understand the physical process, a theoretical analysis was performed, and the details were given in the methods section on the particle self-stabilization model. The results show that the particle self-stabilization is ascribed to the synergy of the following three main factors, as schematically shown in Fig. 2a: (1) At 650 °C, the small wetting angle between YAG:Ce particles and the molten tellurite glass is 43.5°, which promotes the formation of the new interface with an energy barrier of about $6.94 \times 10^5$ zJ, preventing the atomic-scale contact and sintering of YAG:Ce particles in the melt. (2) A attractive van der Waals potential between the YAG:Ce particles in the tellurite glass melt, about $-2.42 \times 10^5$ zJ at the secondary minimum of Fig. 2a, inducing particle aggregation. (3) A weak thermal energy of about 12.74 zJ helps to drive YAG:Ce particle dispersion.

When we pour the YAG:Ce particles into the low-viscosity tellurite glass melt, the YAG:Ce particles are rapidly dispersed in the melt under the agitation drive of the quartz glass rod and the heat drive (Fig. 2b).

Simultaneously, the glass melt quickly wraps the surface of YAG:Ce particles owing to the good wettability between tellurite glass melt and YAG:Ce particles, and forms new interface layers with high energy barrier. The repulsive energy barrier to prevent YAG:Ce particles from contact and sintering, which is the main driving force behind the self-stabilization of YAG particles in the tellurite glass melt, is much higher than attractive van der Waals potential (Fig. 2a). Consequently, the synergy of high repulsive energy barrier and heat energy make the YAG:Ce particles break free from the quasi-clusters formed by van der Waals potential attraction, resulting in dispersed particles in the melt. This particle self-stabilization model allows to create a uniform dispersion of dense particles in liquids and providing more possibilities for functional composites when the repulsive force cannot be generated by conventional techniques. The visualization of intermediate state and final state for YAG: Ce particles in tellurite glass melt by simple agitation, which is helpful to understand the dynamic process of particle dispersion. We utilized the shear stress transport k-omega turbulence model and the discrete phase model in ANSYS Fluent to simulate the flow characteristics of the fluid and the motion state of the particles[38,39]. The particles in the crucible container were dispersed downward in a spiral shape under the different agitation speed to 1, 3, and 5 rev s$^{-1}$ (Fig. 2c and Supplementary Fig. 10). When the agitation

speed is increased to 5 rev s$^{-1}$ for 3–5 s, the particles are uniformly filled throughout the fluid, and the velocity of the particles is almost the same. The simulation results fully demonstrate that the complete dispersion of YAG:Ce particles can be achieved only by using simple agitation strategy under the low viscosity characteristic of tellurite glass.

## Performance characterizations of YAG:Ce-PGC

Thanks for stable particle interface and uniform dispersion of YAG:Ce phosphor particles in tellurite glass, the YAG:Ce-based PGC can be used as an attractive photonic converter. As depicted in Fig. 3a, the photoluminescence excitation (PLE) spectra of YAG:Ce-PGC consist of two excitation bands centered at 343 nm and 450 nm, respectively. Under 450 nm blue light excitation, the $5d \rightarrow 4f$ transition of Ce$^{3+}$ brings a broad yellow emission with the center wavelength of 552 nm[40]. However, the weak excitation peak at 343 nm in YAG:Ce-PGC contributes to the high absorption of the tellurite glass in the ultraviolet range (Fig. 1d). The PLE and PL spectra of YAG:Ce-PGC exhibit similar profiles to YAG:Ce powder, demonstrating that the embedded YAG:Ce particles are the active components of YAG:Ce-PGC. Similarly, the PL decay curves of YAG:Ce-PGC and YAG:Ce powder overlap almost completely (Supplementary Fig. 11). The unchanged steady-state and

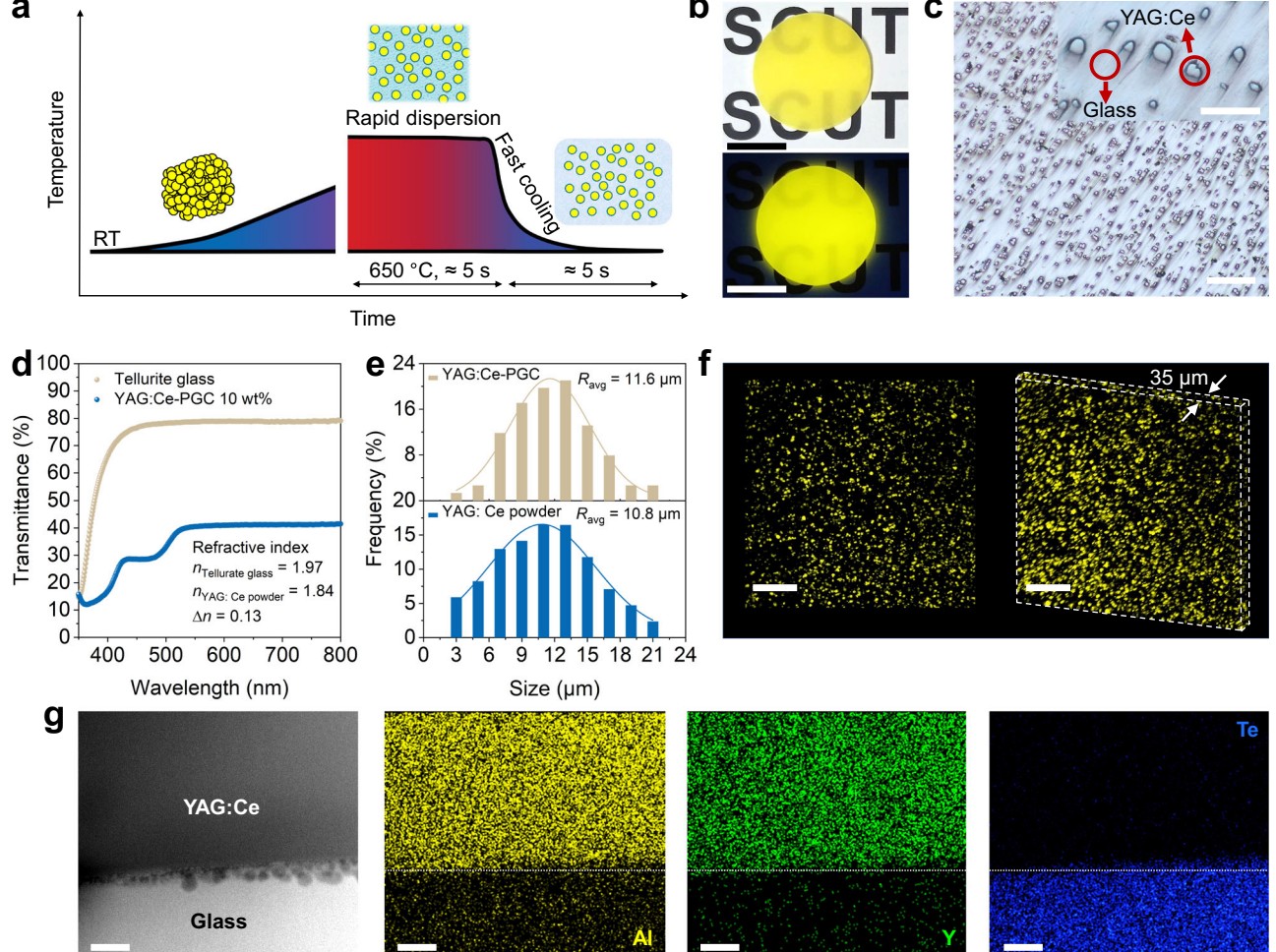

**Fig. 1 | Synthesis and microstructure characterization of phosphor-glass composites. a** Model on temperature-time profile of fast synthesis process. **b** Optical and fluorescence photograph of YAG:Ce-PGC (phosphor content: 10 wt% and also used hereafter.), flat discs sample (diameter of 2.2 cm and thickness of 1.0 mm, scale bar, 1 cm). **c** Optical microscope image of YAG:Ce-PGC (scale bar, 200 μm), inset is enlarged image (scale bar, 50 μm). **d** Transmission spectra of YAG:Ce-PGC 10 wt% and tellurite glass. **e** Comparison of YAG:Ce particle size before and after composite. **f** Surface and 3D reconstruction CLSM image of YAG:Ce-PGC (scale bar, 200 μm). **g** HAADF-STEM image and corresponding EDS mappings profiles across the interface between YAG:Ce crystal and tellurite glass (scale bar, 50 nm). Source data are provided as a Source Data file.

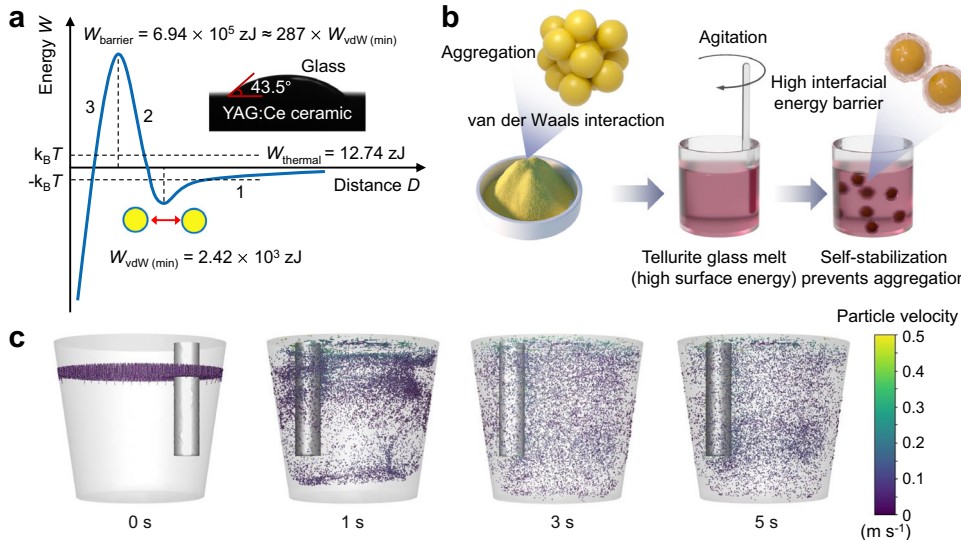

**Fig. 2 | Agitation-based particle self-stabilization model of phosphor-glass composites. a** Potential energy diagram of YAG:Ce particles self-stabilization. $W_{barrier}$ is the interfacial energy barrier between tellurite glass and YAG:Ce particle; $W_{thermal}$ is the thermal energy; $W_{vdW (min)}$ is the van der Waals potential for maximum attraction between two YAG:Ce particles. Segment 1 is attributed to van der Waals interaction potential energy, which induces particle aggregation; segment 2 is dominated by the interfacial energy barrier, which resists the contact of YAG:Ce particles in glass melt; segment 3 is the interfacial energy barrier descent owing to the tellurite glass-YAG:Ce interface being replaced by YAG:Ce interface. The inset is the wetting angle photograph between tellurite glass melt and YAG:Ce ceramics.

**b** Schematic diagram of uniform dispersion and self-stabilization of YAG:Ce particles in tellurite glass melt. The YAG:Ce particles were poured into the low-viscosity and high surface energy tellurite glass melt. The glass melt quickly wrapped the surface of the YAG:Ce particles and formed new interface with high energy barrier, which prevented YAG:Ce particles from contact and sintering based on the self-stabilization model of YAG particles. **c** Time-dependent fluent simulation of YAG particles dispersed in tellurite glass melt (at different stages of 0, 1, 3 and 5 s) under the agitation speed of 5 revolutions per second (rev s⁻¹), and the color depth of the particle represents the velocity at that moment. Images used courtesy of ANSYS, Inc.

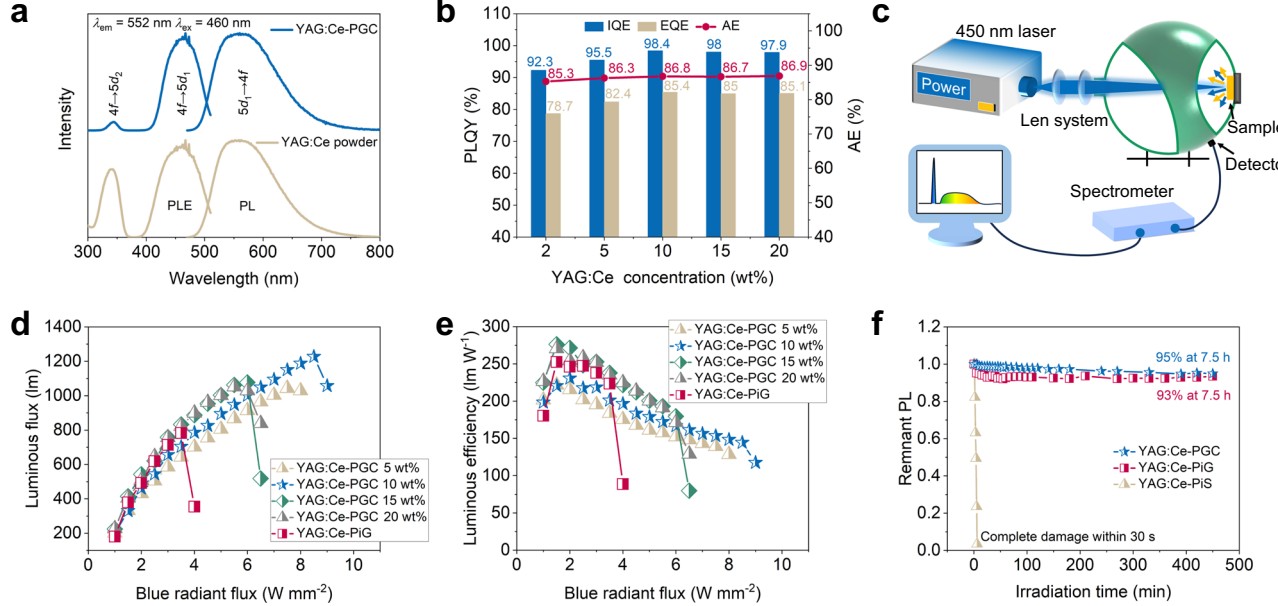

**Fig. 3 | Optical characterization of YAG:Ce-PGC flat discs. a** PLE and PL spectra of YAG:Ce-PGC and YAG:Ce powder. **b** Internal/external quantum efficiency (I/EQE) and absorption efficiency (AE) of YAG:Ce-PGC. IQE (AE) of YAG:Ce powder was determined to be 99.4% (74.5%). **c** Schematic diagram of laser driven light source test system. A 450 nm blue laser is emitted from the laser source and irradiated onto the sample using two optical focusing mirror. The light emitted by the sample and the reflected blue laser are then collected using an integrating sphere and analyzed within the fiber optic spectrometer. **d** Luminous flux and **e** the luminous efficacy of YAG:Ce-PGC with different concentrations as a function of the laser power density. **f** Photostability of the YAG:Ce-PGC, commercial YAG:Ce phosphor in glass (YAG:Ce-PiG), and YAG:Ce phosphor in silicone resin (YAG:Ce-PiS) under 2 W mm⁻² blue laser irradiation. Source data are provided as a Source Data file.

transient PL properties of YAG:Ce further confirm the negligible interface reaction between YAG:Ce and tellurite glass. As portrayed in Fig. 3b, the IQE of all samples is no less than 92%, with an absorption of no less than 85% (Supplementary Table 2). The high quantum efficiency and absorption are put down to the fact that the undamaged YAG:Ce particles are densely and uniformly dispersed in the tellurite glass matrix, producing high-efficient light scattering. Moreover, YAG:Ce-PGC 20 wt% sample possesses thermal conductivities of 1.52

and 1.78 W m$^{-1}$ K$^{-1}$ at ambient temperature and 250 °C, respectively, which is seven times that of organic resins (0.2 W m$^{-1}$ K$^{-1}$) and comparable to silica glass (1.49 W m$^{-1}$ K$^{-1}$) (Supplementary Fig. 12)[25]. The relatively high thermal conductivity also facilitates the application of high-power light sources. The proposed strategy is a general way for the development of various PGC. To verify this, diverse translucent PGC samples have been synthesized by replacing YAG:Ce with LuAG:Ce and GdAG:Ce (denoted as LuAG:Ce-PGC and GdAG:Ce-PGC, respectively). As expected, all the as-synthesized PGC are highly efficient with limited IQE loss (Supplementary Fig. 13).

To evaluate the potential application of laser-driven white light illumination, the photoelectric properties of YAG:Ce-PGC were characterized (Fig. 3c). Under the excitation of focused blue laser, the luminous flux (LF) initially showed a linear increase with the incident laser power, reaching its peak value and then sharply decreased (Fig. 3d). This emission saturation at high power density is commonly observed in laser-driven illumination, attributed to the accumulated heat in the color converter and subsequent PL thermal quenching[33]. With increasing doping concentration, the saturation threshold of the YAG:Ce-PGC sample first increased and then decreased, almost the same as that of the sample temperature (Supplementary Fig. 14), which indicates the higher the concentration of YAG:Ce phosphor, the more blue lasers may be down-converted in the same radiation region, generating more heat and accelerating the luminous saturation of the sample[26]. Prior to the occurrence of emission saturation, the luminous efficiency gradually decreased (Fig. 3e), and the maximum values of luminous flux, luminous efficiency, and the laser saturation threshold are 1227 lm, 276 lm W$^{-1}$ mm$^{-1}$, 8.5 W mm$^{-1}$, respectively. It is found that the as-measured parameters are superior to commercial YAG:Ce phosphor in glass (YAG:Ce-PiG), and the detailed composition is shown in Supplementary Fig. 15 and Supplementary Table 3. The corresponding CRI and CCT decreased with increasing doping concentration (Supplementary Fig. 16). The CRI and CCT of the PGC samples with YAG:Ce doping concentration between 10 wt% and 15 wt% show a sudden drop, attributed to the synergy of increase in YAG:Ce content and porous characters (Supplementary Fig. 17). The thermal performance of the color converter is crucial in light source applications. As shown in the supplementary Fig. 18, the integrated emission intensity of YAG:Ce-PGC at 150 °C can still maintain 96% of that at 30 °C, and is also superior to that of YAG:Ce phosphor in silicone resin (YAG:Ce-PiS) and YAG:Ce powder. In addition, the operational stability of composites was tested under different encapsulation strategies under continuous 2 W mm$^{-2}$ blue laser irradiation. As plotted in Fig. 3f, YAG:Ce-PGC exhibited comparable operation stability as the commercial YAG:Ce-PiG under continuous 7.5 h blue laser irradiation, while the YAG:Ce-PiS was irreversibly destroyed merely within 30 s at the same power. These excellent optical performance and stability undoubtedly validate that the rapid synthesis strategy of YAG:Ce fully encapsulated in dense tellurite glass can not only achieve phosphor particle integrity, but also effectively protect YAG:Ce phosphor from oxygen, moisture, light exposure, and heat damage.

In summary, we propose a facile and fast agitation synthetic protocol in seconds for phosphor-glass composites with stable particle interface and dense uniform dispersion. Theoretical and experimental studies reveal that the dense uniform dispersion of phosphor particles in tellurite glass originates from the good wettability between YAG:Ce particles and the tellurite glass melt, which creates an energy barrier to prevent particle clustering in the melt. The nondestructive interface of phosphor and tellurite glass can be emanated in local structures and confirmed by the improved luminescence properties. The resultant YAG:Ce-based PGC possess high quantum efficiency of 98.4% and absorption coefficient of 86.8%, presenting great potential for high-power white lighting. In addition, the universality of the strategy is confirmed, and different color-tunable translucent composites are also prepared. This work provides a straightforward and general synthetic

protocol to functional glass composites for various light-emitting and light-detecting applications, and further expands the scope of imagination for the design and practical application of glass based composite materials with high stability.

## Methods

### Raw materials
TeO$_2$, ZnO$_2$, and Na$_2$CO$_3$ (Shanghai Aladdin Biochemical Technology Co., Ltd.) were used as the starting ingredients with a purity of 99.99%. Commercial yellow phosphor YAG:Ce, green phosphor LuAG:Ce, orange-yellow phosphor GdAG:Ce, and silicone resin A/B was purchased from Shenzhen Looking Long Technology Co., Ltd, China. Commercially available YAG:Ce-PiG was offered by Bright Phosphor Composites Tech., Ltd. in China.

### Precursor glass, YAG:Ce-PGC, and YAG:Ce-PiS fabrications
The precursor tellurite glass matrix consisted of 75TeO$_2$, 15ZnO, and 10Na$_2$O (mol%) was produced using the traditional melt-quenching method. The process involved melting the corresponding raw ingredients in the presence of air for 40 min at 700 °C and subsequent melt-quenching to form glass. The as-prepared precursor glass was then crushed and ground into powders for the second melting process. The glass powder is put into the corundum crucible and melted in 650 °C for 40 min until the high-temperature melt is completely clarified. The crucible is removed and the YAG:Ce phosphor is quickly poured into the high-temperature melt, and the quartz glass agitation rod is quickly agitated for about 5 s, then the melt mixture is immediately poured out to form and placed in the 280 °C annealing furnace to remove internal stress. YAG:Ce phosphor in silicone resin (YAG:Ce-PiS) sample with 10 wt% phosphor concentrations was prepared in this study for a comparison. The YAG:Ce-PiS sample preparation method is as follows: the silicone resin A/B and phosphor are first weighed according to the mass ratio, stirred evenly, and then subjected to vacuum drying to remove the bubbles. Finally, the mixture is injected into the disc mold, heated to 100 °C for 3 h, and finally the composite material is obtained.

### Characterizations
An Aeris X-ray diffractometer (PANalytical Corp.) operating at 40 kV and 15 mA with monochromatized Cu Kα radiation (λ = 1.5406) and a linear VANTEC detector was employed for collecting the powder X-ray diffraction (XRD) patterns. Using a confocal laser scanning microscope (TCS SPE, Leica, Germany) at an excitation wavelength of 488 nm, the 3D distribution of phosphor particles within tellurite glass was studied. PL spectra and decay curves were characterized using a high-resolution spectrofluorometer (FLS1000, Edinburgh Instruments, UK) equipped with a Xe lamp as an excitation source and R928 photomultiplier (200−900 nm) as a detector. The temperature-dependent emission spectra were measured on a temperature controller in the same instrument. The Raman spectra were recorded on a Raman spectrometer (Renishaw in Via, London, UK) under the excitation of a 785 nm laser. The QE and absorption efficiency of all samples were measured by the absolute photoluminescence quantum yield spectrometer (Hamamatsu, C13534). The morphology image of YAG:Ce was acquired on SEM (Zeiss Merlin), and, and a layer of gold was sputtered to the surface of the sample to enhance the conductivity. The surface morphology and compositions of the samples were characterized by a TEM (Titan Themis Z 300 kV, Thermos-Fisher scientific) and a dual-beam FIB microscope (FIB-SEM, Helios G4, Thermo Scientific, USA) was used to thin the sample. High-resolution lattice fringe and selected area electron diffraction (SAED) were tested using environmental transmission electron microscopy (FEI Titan G2 60−300). The thermogravimetric-differential scanning calorimetric (TG-DSC) curves of precursor glass powders were detected by a simultaneous thermal analyzer (STA, STA449C, NETZSCH, Germany). A heating rate of 10 °C min$^{-1}$ and air atmosphere were adopted. The surface tension and

was measured by a surface tension measuring instrument (Dataphysics OCA25-HTV1800, Germany). The viscosity curve of tellurite glass was tested by a high temperature viscometer (BCT1700, Shanghai Huanao Technology Co., LTD). Using a Metricon 2010/M prism coupler, the refractive indexes of tellurite glass were measured. The dielectric constant of samples is measured by a precision impedance analyzer (6500B/4294A WAYNE KERR/Agilent). Thermal diffusivities of the PGC were measured by a laser flash apparatus (LFA 467, Netzsch, Germany).

### Fluent simulation model description

The geometry of the model is shown in Supplementary Fig. 10a. The quartz glass agitation rod has a diameter of 5 mm. The composition of liquid tellurite glass is $75TeO_2–15ZnO–10Na_2O$, which has a density of $5020 \ kg \ m^{-3}$ and a viscosity of $0.08693 \ kg \ ms^{-1}$. The YAG:Ce particles with an average particle size of 10.8 μm and density of $4700 \ kg \ m^{-3}$ are treated as inert-particles. The 10 wt% particles can be injected above the top of the crucible for 1 s. We utilized the shear stress transport k-omega turbulence model and the discrete phase model in ANSYS Fluent to simulate the flow characteristics of the fluid and the motion state of the particles. This model enabled transient simulation and incorporated gravity to investigate the intricate interactions between molten tellurite and YAG:Ce particles.

### YAG:Ce particles self-stabilization model

The self-stabilization of YAG:Ce particles in tellurite glass melt is contributed to the synergistic effect of attractive van der Waals forces between YAG:Ce particles, thermal energy, and a high interfacial energy barrier due to a good wettability between YAG:Ce particles and tellurite glass melt. The interfacial energy barrier is much higher than the van der Waals attraction and thermal energy, so it can prevent the atomic-size contact and sintering of YAG:Ce particles, breaking the limits of agglomerations in tellurite glass melt, as shown in Fig. 2a, b.

**van der Waals attraction.** The van der Waals interaction for two YAG:Ce particles in tellurite glass melt at 923 K can be approximately calculated by the following equation:[41–43]

$$W_{vdW}(D) = -\frac{A(a^2 + RD)}{12D^2} \quad (1)$$

$$A = \frac{3}{4} k_B T \left( \frac{\varepsilon_G - \varepsilon_Y}{\varepsilon_G + \varepsilon_Y} \right)^2 + \frac{3h\nu_e}{16\sqrt{2}} \frac{(n_G^2 - n_Y^2)^2}{(n_G^2 + n_Y^2)\sqrt{n_G^2 + n_Y^2}} \quad (2)$$

where $D$ is the distance between two particles in nanometers. $A$ is the Hamaker constants for the van der Waals interaction. $a$ is the radius of the particle contact surface ($a$ is negligible relative to the micron size $R$ of the YAG:Ce particles). $R$ is the radii of YAG:Ce particles. $k_B$ is the Boltzmann constant; $T$ is the ambient temperature; $\varepsilon_G$ and $\varepsilon_Y$ are the relative permittivity of tellurite glass and YAG:Ce, respectively. h is Planck constant. $\nu_e$ is the main electron absorption frequency in ultraviolet light. $n_G$ and $n_Y$ are the refractive indices of tellurite glass and YAG:Ce, respectively. According to the parameters in Supplementary Table 1, it can be calculated that $A$ is approximately equal to 2.15 zJ. Therefore, the van der Waals interaction between two similar YAG:Ce particles in tellurite glass melt can be further derived as follows:

$$W_{vdW}(D) = -\frac{2.15R}{12D} \quad (3)$$

Equation (3) is valid only if the contact distance $D$ between two YAG:Ce particles in tellurite glass melt is about more than two atomic layers (about 0.4 nm)[44,45]. Thus, when $D = 0.4$ nm, the maximum attraction $W_{vdW \ (min)}$ between two YAG:Ce particles in tellurite glass

melt is calculated to be $-2.42 \times 10^3$ zJ. The van der Waals potential energy easily induces YAG:Ce particles to aggregate when the other repulsive force cannot be generated.

**Thermal energy for particle dispersion.** The thermal energy of YAG:Ce particles for Brownian motion, $E_b$, can be estimated by[44]

$$E_b = k_B T \quad (4)$$

where $T$ is the absolute temperature. At a synthesis temperature of 923 K, the $E_b$ value of 12.74 zJ is less than the maximum van der Waals attraction in tellurite glass-YAG system, but it also contributes to the dispersion of YAG:Ce particles when the distance $D$ between two particles is relatively far.

**Interfacial energy barrier preventing particle contacting and sintering.** In the tellurite glass-YAG:Ce system, when the two YAG:Ce particles are close to each other at a distance of about 0.2 nm, the last atomic layer will be extruded, and the particle contact may induce sintered particles. The tellurite glass-YAG:Ce interface will be replaced by a YAG:Ce surface. Therefore, we can calculate the interface energy barrier to estimate the possibility of two particles being sintered. The interfacial energy barrier will be calculated by:

$$W_{barrier} = S(\sigma_Y - \sigma_{Y-G}) \quad (5)$$

According to Young's equation, $W_{barrier}$ will be calculated by:

$$W_{barrier} = S\sigma_G \cos\theta \quad (6)$$

where $S$ is the effective area, $\sigma_Y$ is the surface energy of YAG:Ce, $\sigma_{Y-G}$ is the interfacial energy between YAG:Ce and tellurite glass melt, $\sigma_G$ is the surface energy of tellurite melt, and $\theta$ is the contact angle of tellurite glass melt on YAG:Ce surface. The equation clearly shows that the better the wettability (the smaller the $\theta$) between the particles and the molten glass in this synthesis method, the higher the interfacial energy barrier that prevents the particles from contacting each other.

The surface energy of tellurite glass melt is $0.141 \ J \ m^{-2}$ (Supplementary Table 1). As shown in Fig. 2a, at 923 K, the contact angle between the tellurite glass and YAG:Ce is 43.5°. According to the Lambert approximation, ideally the effective interaction area between the two spheres is $S = 2\pi R D_0$ (where $D_0 = 0.2$ nm)[45]. For two YAG:Ce particles with a radius of 5.4 μm, the interface barrier is calculated to be $6.94 \times 10^5$ zJ, which is more than 280 times the van der Waals potential energy that causes the particles to cluster. Therefore, it is almost impossible for YAG:Ce particles to overcome the interfacial energy barrier to contact each other for sintering. Thus, YAG:Ce particles can still exist stably after being dispersed in tellurite glass melt.

### Reporting summary

Further information on research design is available in the Nature Portfolio Reporting Summary linked to this article.

## Data availability

The data that support the findings of this study are available from the corresponding author upon request. Source data are provided with this paper.

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

## Acknowledgements

This research was supported by National Natural Science Foundations of China (grant no. 51972118), and the Local Innovative and Research Teams Project of Guangdong Pearl River Talents Program (2017BT01X137).

## Author contributions

X.Z.G. conceived the initial concept. Y.S.S. prepared the sample and processed the experimental data. X.Z.G., Y.S.S., Y. Z. W., W. B. C., Q. Q. J., D. D. C and G. P. D. interpreted the theoretical and experimental results. Y.S.S. wrote the paper, X.Z.G supervised the work and revised the paper. All authors discussed and edited the paper.

## Competing interests

The authors declare no competing interests.
