## [Peer Review File · Nature Communications]

Rapid synthesis of phosphor-glass composites in seconds based on particle self-stabilizationEditorial Note: Parts of this Peer Review File have been redacted as indicated to remove third-party material where no permission to publish could be obtained.

REVIEWER COMMENTS

Reviewer #1 (Remarks to the Author):

The authors report on the rapid synthesis of phosphor-glass composites. This is a nice and original piece of work which, to me, deserves publication. The work is technically sound and the authors demonstrate their findings. However I would like a few points to be discussed and possibly detailed before recommending acceptance. They are listed below:

- Could the authors discuss, and possibly model, the transparency of their materials? YAG is isotropic (cubic) but its refractive index does not exactly match the one of the tellurite glass used. I think that they should be able to model the transmittance of their material using the amount of YAG used as well as the size of the particles.

- Did the authors try to modify the glass refractive index to match better the value of YAG? If so what happened? If not why? I guess that there is room for improvement regarding the transparency here. Decreasing the glass refractive index by modifying its composition would enable a better refractive index match and maybe would enable to increase the amount of YAG which can be inserted in the final material.

- If the authors do not wish to modify the glass, maybe choosing a garnet material with a higher refractive index could help to achieve a better transparency. LuAG could be an interesting candidate.

- The authors do not clearly state the amount of YAG in the sample used for transmittance measurement presented figure 1. Could they provide more details please? Also a figure (could be placed in SI section) showing the different transmittance curves as a function of the YAG content should be shown.

- Did the authors try to use YAG nanocrystals to avoid light scattering? It would be interesting to see if the nanoparticles segregate.

- Could the authors discuss what the important factors are to choose a glass matrix which will enable such dispersion with different crystals?

I hope these comments can help to further improve the quality of the manuscript.

Congratulations for your nice work.

Reviewer #2 (Remarks to the Author):

The authors of this paper report on the rapid synthesis and characterization of the phosphor-glass composites, which were composed of YAG:Ce and tellurite glasses. It is technologically important and will open the way for the fabrication of phosphors-in-glass (PiG) composites. The experiments were well done on the optical properties of the obtained composites in comparison with those of raw YAG:Ce phosphor. No chemical reactivity of YAG:Ce powder and 75TeO₂-15ZnO-5Na₂O tellurite glass was demonstrated by not only the optical estimations but also TEM observation and EDS analysis, as well. In addition, the mixing process of the particle dispersion was investigated by the simulation with good visualization as seen in Fig.2, which seemed to be performed with the parameters obtained by using the agitation-based particle self-stabilization model. However, the present reviewers cannot recommend this paper for publication because it has the following problems and questionable points. We hope that our comment is useful for the improvement of the paper.

(1) Generally, the word "tellurate" means a compound containing an oxyanion of tellurium having an oxidation number of +6. Because the glass used in this paper is a TeO₂-based glass, the word "tellurate" should not be used.

(2) In the main text, there are no descriptions for abbreviations for PiG and PiS. In addition, information for YAG:Ce-PiG and -PiS is insufficient. What is a component of a matrix glass of the former? How did the authors obtain the latter? In the caption of Fig. 3, the abbreviation of "phosphor in silicon" is written as PiS, but does that mean "phosphor in silicone"?

(3) Regarding Fig. 3d and 3e, the authors mentioned that the saturation threshold firstly increased and then decreased with increasing the concentration, possibly due to an increase of porosity resulting in a reduction of thermal conductivities and mechanical strengths. The authors should show evidence for them.

(4) In abstract, they mentioned "we develop a rapid synthetic route to PGC within ~ 10sec" and in maintext, "to achieve a dense uniform dispersion of YAG:Ce phosphor particles in tellurate glass within 5 s". Which is insisted, 10 or 5 sec?

(5) The temperature rise under blue laser excitation should be monitored.

(6) It is understandable that the choice of tellurite glasses among various glass systems (silicate, borate, phosphate etc.) is good, as seen in Supplementary Fig.1b. But there was not provided information on why the 75TeO₂-15ZnO-5Na₂O composition was chosen.

(7) Page 4, line 49. It is seen to have a simple mistyping of Fig.1g in "YAG:Ce crystals and ... (Fig.1f)."

(8) Page 5, line 2-3, ".. back into the high-temperature furnace to continue heating and ..". Please identify the temperature used here.

(9) Page 5, line 10-12. "At 650°C, the small wetting angle .., preventing the atomic-scale contact and sintering of YAG:Ce particles in the melt." Why can the high wettability (small contact angle 43.5°) prevent the atomic-scale contact? The wettability of YAG:Ce with the tellurite glass surely gave higher dispersion of YAG:Ce particles in the tellurite glasses (Their description in Page 11, line 19-20, "the better wettability (the smaller the theta) between the particles and the molten glass in this synthesis method, the higher the interfacial energy barrier that prevent the particles from contacting each other" is correct), but that does always not mean less reactivity for YAG:Ce particles and the molten glass. The interfacial energy (barrier) will not be the barrier energy of the reaction between them.

(10) How were the experimental parameters of barrier energy, thermal energy and attractive van der Waals energy used in the simulation found in Fig.2c? The reviewer noticed that they mentioned "the YAG:Ce particles are treated as inert-particles." If so, the simulation presented (Fig.2c) was not any imitation of the mixing behavior with melt-particle interaction (wettability) and van der Waals interaction working between YAG:Ce particles suggested in Fig.2a.

(11) No detail on the information about YAG:Ce-PiSG (silica glass?) was provided, either. Was ref.4 referred in Supplementary Fig.11 correct? Ref.25 showed that the silica glass in YAG:Ce-PiSG was derived from triblock co-polymer and TEOS and subsequently sintered with YAG:Ce powders by SPS technique. Thus, the silica glass was not a quartz glass (melt-quenched) but a sol-gel derived silica glass. Anyway, the manuscript is difficult to be read.

(12) The authors explained, for example in page 8 line 26-27, "the rapid synthesis strategy of YAG:Ce encapsulated in dense tellurate glass can not only achieve phosphor particle integrity, ..", while their observation was "the saturation threshold of the YAG:Ce-PGC sample first increased and then decreased, possibly due to the increased porosity in the high-doping YAG:Ce-PGC samples." They seemed to mention that the samples were dense and porous, which was difficult to be understood: The latter observation was true for the 10wt% doping as well as for 20wt% doping, as seen in Fig.3d. That means the 10wt% doped PGC sample also had higher porosity and degraded thermal conductivity, if the authors' claim was correct. It is contradiction and confusing.

(13) Please more clearly identify the optimized concentration of YAG:Ce powders for the fabrication of YAG:Ce-PGCs. It must be 10wt%.

(14) They mentioned "the corresponding CRI and CCT decreased with increasing doping concentration, mainly due to the increase in yellow components (Supplementary Fig.13)." To see it more closely, the 5 and 10wt% concentrations gave almost the same properties but the 15wt% decreased the CRI and CCT in comparison with the lower concentrations. These behavior was almost identical for the 20wt% concentration, as well. Careful description would be required.

(15) In Fig.3d, a part of data was hidden by the legend of the figure. Please revise it.

Reviewer #3 (Remarks to the Author):

Reviewer #4 (Remarks to the Author):

The manuscript „Rapid synthesis of phosphor-glass composites in seconds“ presents an interesting experimental investigation of YAG:Ce particles uniformly embedded in a glass matrix. The authors have carried out extensive and careful experimental work and meaningful analytical studies of the samples, taking into account the relevant literature and discussing the results with theoretical models. The extensive appendix supports the understanding and interpretation of the results.

The work has practical relevance and may help to solve the problem of degradation of phosphors at high irradiation energy.

The manuscript is recommended for publication in Nature Communications with the following minor corrections:

Pg. 4 : the numerical value of the refractive index tellurate glass (1.82) and YAG:Ce crystal (1.97) was exchanged

Fig. 2a: the visibility of the wetting angle in the graphic is poor

Fig. 2c: use r/s instead of s

Pg. 5: "to continue heating".. at 650°C?

Fig. 3f: delete ", and YAG:Ce phosphor in glass," in the caption

Reviewer #1 (Remarks to the Author)

Comment. The authors report on the rapid synthesis of phosphor-glass composites. This is a nice and original piece of work which, to me, deserves publication. The work is technically sound and the authors demonstrate their findings. However I would like a few points to be discussed and possibly detailed before recommending acceptance. They are listed below:

Response. We thank the Reviewer for the positive comments. According to your suggestions, we have made relevant revisions. The revised sentences were highlighted with blue font color in the revised manuscript. Hope it will be accepted after this revision.

Question 1. Could the authors discuss, and possibly model, the transparency of their materials? YAG is isotrope (cubic) but its refractive index does not exactly match the one of the tellurite glass used. I think that they should be able to model the transmittance of their material using the amount of YAG used as well as the size of the particles.

Author reply: Thanks for the suggestion and informative comments. Yes, we try to discuss it in this revision. According to the relationship between the scattered particle radius (d) and the incident light wavelength (λ), the widely recognized and adopted theoretical models include Rayleigh scattering and Mie scattering; and their application ranges are shown in Fig. R1a. Mie and P. Debye built a model and calculated with spherical particles that, when $d < 0.3\lambda/2\pi$, Rayleigh scattering is obeyed between scattered light and incident light (Ref: *Ann. Phys. Berlin*, 330, 377-445 (1908)). To simplify calculations, it is generally believed that the radius d of the scattered particle is less than 1/10 of the incident wavelength, that is, $d < 0.1\lambda$, it can also be seen as following Rayleigh scattering. When the scattering particle radius d is close to the incident wavelength λ (generally believed to be $d/\lambda \approx 0.1-8$), Mie scattering is obeyed.

In order to verify the feasibility, we attempt Mie scattering model to calculate the relationship between crystal doping concentration and grain size, and transmittance (Mie scattering calculation process by MATLAB. Ref.: *IAP Res. Rep.* 8, 9 (2002)). When the doping content is only 2% and the particle size is above 1 μm , the transmittance of the composite material begins to decline rapidly (Fig. R1b). However, the particles' size of YAG:Ce in our work is about 11 μm (It is suitable for the lighting applications with high luminescence efficiency), which is too large to apply to the Mie scattering model (suitable for scattering conditions with a similar size between the crystal size and the incident light wavelength), leading to considerable errors. Moreover, we also tried the Fraunhofer diffraction model and the Apetz-van Bruggen model (Ref: *J. Am. Ceram. Soc.* 86 480-86 (2003); *J. Eur. Ceram. Soc.* 41, 2169-2192 (2021).) to calculate the transmittance of our composite materials, but there are also considerable errors due to same problem. Meanwhile, the COMSOL simulation was considered to calculate the transmittance of the composite material, but due to the large amount of calculation, the final result could not be obtained. Therefore, the construction of the calculation model for the permeability of this large-size particle composite is still a thorny problem. We also want to point out that phosphor particles with big size ($\sim 10 \mu\text{m}$) used in this phosphor-glass composites benefit to the enhanced luminescence efficiency for lighting applications.

Fig. R1 a Rayleigh scattering and Mie scattering model. **b** Relationship between crystal size and doping concentration and the theoretical transmittance of phosphor glass composites.

Question 2. Did the authors try to modify the glass refractive index to match better the value of YAG? If so what happened? If not why? I guess that there is room for improvement regarding the transparency here. Decreasing the glass refractive index by modifying its composition would enable a better refractive index match and maybe would enable to increase the amount of YAG which can be inserted in the final material.

Author reply: Yes, we agree that modifying the glass refractive index to match better the value of YAG for improvement regarding transparency and increasing the amount of YAG:Ce is feasible. The refractive index (n) of the tellurite glasses families varies between 1.93 and 2.30 (Ref.: *Prog. Mater. Sci.* 57, 1426-1491 (2012)). The tellurite glass consisting of 75 TeO₂-15 ZnO-10 Na₂O ($n = 1.97$) in our manuscript has a good capacity for YAG:Ce phosphor, and the quantum efficiency of YAG:Ce-PGC 10 wt% sample has reached the highest value (Abs = 86.8%, IQE = 98.4%) (Fig. 3b), and possesses a decent transmittance of about 40%. Therefore, on the premise of ensuring luminous performance and transmittance, our PGC samples possess a relatively high phosphor content. However, for YAG:Ce-PGC materials obtained by the rapid synthesis method in seconds, the factors are not only the matching of refractive index but also the **index of thermal stability** ΔT ($\Delta T = T_x - T_g$; the larger the ΔT , the stronger the resistance to crystallization of glass), **melting point**, and **viscosity** of the glass system, which are very important to improve the dispersivity of phosphor and reduce thermal erosion. The melting temperature of glass is related to the polarizability of its constituent cations. The higher the cation polarizability, the lower the melting temperature of the glass (Ref: *Proc. R. Soc. Lond. A* 217, 203-221 (1953)). Cations in the outer layer that contain non-inert electron pairs (such as Pb²⁺, Bi³⁺, Te³⁺, etc.) and an 18-electron configuration (such as Zn²⁺) have higher polarizability (Ref: *Prog. Mater. Sci.* 57, 1426-1491 (2012)). In addition, the introduction of ZnO is beneficial to neutralize the covalency of Te-O bond and improve the glass formation ability (Ref.: *J. Non-Cryst. Solids* 151, 134-142 (2021)). The addition of Na₂O can further reduce the melting temperature and viscosity, drive the triangular cone [TeO₄] in the glass network to [TeO₃], and improve the thermal stability ΔT of the glass (Ref.: *J. Alloy. Compd.* 854, 157072 (2021)). Therefore, we studied the TeO₂-ZnO-Na₂O glass system, and the composition of 75 TeO₂-15 ZnO-10 Na₂O was screened and selected (Fig. R2 and Table

R1).

Fig. R2 | a-b DSC curve of $\text{TeO}_2\text{-ZnO-Na}_2\text{O}$ (T-Z-N) glasses.

Table R1 Composition and thermal properties of $\text{TeO}_2\text{-ZnO-Na}_2\text{O}$ glasses.

Element	T_g (°C)	T_x (°C)	ΔT (°C)
75 $\text{TeO}_2\text{-20ZnO-5Na}_2\text{O}$	308	349	41
75$\text{TeO}_2\text{-15ZnO-10Na}_2\text{O}$	279	347	69
75 $\text{TeO}_2\text{-10ZnO-15Na}_2\text{O}$	269	330	61
70 $\text{TeO}_2\text{-25ZnO-5Na}_2\text{O}$	315	365	50
70 $\text{TeO}_2\text{-20ZnO-10Na}_2\text{O}$	291	344	53
70 $\text{TeO}_2\text{-15ZnO-15Na}_2\text{O}$	271	332	61
70 $\text{TeO}_2\text{-10ZnO-20Na}_2\text{O}$	251	306	55

Question 3. If the authors do not wish to modify the glass, maybe choosing a garnet material with a higher refractive index could help to achieve a better transparency. LuAG could be an interesting candidate.

Author reply: Very nice suggestion! Yes, LuAG:Ce material with higher refractive index can indeed achieve better refractive index matching and increase the doping content of phosphor. However, as a blue light excited white light source, LuAG:Ce phosphor shows the main green emission, leading to a low color rendering index (CRI < 55) that is not conducive to white light application (**Ref.:** *Laser Photonics Rev.* 6, 2200909 (2023); *Laser Photonics Rev.* 12, 2200553 (2023)). In addition, LuAG:Ce phosphor has a price four times higher than YAG:Ce phosphor (Bright Phosphor Composites Technology, China) (<http://www.bpc-t.cn/PR2/13.html>). Therefore, YAG:Ce phosphor was chosen as a proof of concept for our experiment after comprehensive consideration in the applications. We have also verified the feasibility of LuAG:Ce-PGC and GdAG:Ce-PGC in the supplementary information (Supplementary Fig. 13), and it also exhibits high quantum efficiency and interesting dispersion of phosphor particles. Thank you for your suggestion. We think it will be a meaningful topic in future research.

Question 4. The authors do not clearly state the amount of YAG in the sample used for transmittance measurement presented figure 1. Could they provide more details please? Also a figure (could be placed in SI section) showing the different transmittance curves as a function of the YAG content should be shown.

Author reply: Thanks for your suggestion. We have provided more details on the mentioned contents. We have firstly supplemented that the YAG:Ce phosphor content of YAG:Ce-PGC samples in transmittance measurements is 10 wt%. Moreover, we have also added the transmittance spectra of YAG:Ce-PGC samples with different phosphor content, as shown in Fig. R3 (Supplementary Fig. 3 in the revised version).

Fig. R3 | Transmission spectra of phosphor-glass composites with different YAG:Ce content.

Question 5. Did the authors try to use YAG nanocrystals to avoid light scattering? It would be interesting to see if the nanoparticles segregate.

Author reply: It is true that YAG nanocrystals can indeed reduce the occurrence of light scattering; however, with the decrease in crystal size (especially $< 1 \mu\text{m}$), their quantum efficiency of the phosphors decreases sharply because of low crystallinity, abundant surface defects, etc., which, from the point of view of the source of energy, makes them unsuitable for use as a lighting source (Ref: *Chem. Rev.* 120, 13461–13479 (2020)). Therefore, we have not made this kind of attempt, but if the nanocrystal-glass composites with good refractive index matching can be achieved through the rapid synthesis method in seconds, it will show more interesting phenomena, which will be another meaningful research topic.

Question 6. Could the authors discuss what the important factors are to choose a glass matrix which will enable such dispersion with different crystals?

Author reply: The important factors in the selection of glass matrix for phosphor-glass composites prepared through rapid synthesis route are discussed as follows:

1. Regarding the interfacial energy barrier (equation (6) in our manuscript), the equation clearly shows that the better the wettability (the smaller the contact angle θ) between the particles and the molten glass, the higher the interfacial energy barrier that prevents the particles from contacting each other. Thus, the wettability between glass melts and crystal particles is one of the most critical factors. Equally, select a glass system with a high melt

surface energy to improve the solid-liquid interface energy barrier and prevent particles from agglomerating.

2. Regarding the van der Waals attraction of crystal particles (equations (1-2) in our manuscript), Hamaker constants A , depending on dielectric constant and refractive index n between crystal particles and glass matrix, are proportional to the van der Waals attraction of crystal particles in glass melt medium. Obviously, similar dielectric constants and refractive indexes between crystal particles and glass matrix will lead to a small A value, reducing the inter-particle van der Waals attraction energy.

3. For composite materials obtained by the rapid synthesis method in seconds, the factors are not only the matching of refractive index but also the index of thermal stability ΔT ($\Delta T = T_x - T_g$, the larger the ΔT , the stronger the resistance to crystallization of glass), melting point, and viscosity of the glass system, which are also very important to improve the dispersivity of phosphor and reduce thermal erosion (also answered in question 2).

Reviewer comment: I hope these comments can help to further improve the quality of the manuscript. Congratulations for your nice work.

Author reply: Thank you again for your positive comments about our manuscript. It will receive broad interest for the interesting results.

Finally, we thank you for the constructive and valuable comments again.

Reviewer #2 (Remarks to the Author):

Comments. The authors of this paper report on the rapid synthesis and characterization of the phosphor-glass composites, which were composed of YAG:Ce and tellurite glasses. It is technologically important and will open the way for the fabrication of phosphors-in-glass (PiG) composites. The experiments were well done on the optical properties of the obtained composites in comparison with those of raw YAG:Ce phosphor. No chemical reactivity of YAG:Ce powder and 75TeO₂-15ZnO-5Na₂O tellurite glass was demonstrated by not only the optical estimations but also TEM observation and EDS analysis, as well. In addition, the mixing process of the particle dispersion was investigated by the simulation with good visualization as seen in Fig.2, which seemed to be performed with the parameters obtained by using the agitation-based particle self-stabilization model. However, the present reviewers cannot recommend this paper for publication because it has the following problems and questionable points. We hope that our comment is useful for the improvement of the paper.

Response. We thank this Reviewer for the valuable comments on our manuscript. Yes, as the Reviewer summarized, "it is technologically important and will open the way for the fabrication of phosphors-in-glass (PiG) composites." We also appreciate the problems and questionable points proposed by this Reviewer, and we have made relevant revisions according to the suggestions. The revised sentences were highlighted with blue font color in the revised manuscript.

Question 1. Generally, the word "tellurate" means a compound containing an oxyanion of tellurium having an oxidation number of +6. Because the glass used in this paper is a TeO₂-based glass, the word "tellurate" should not be used.

Author reply: Yes, this is perfectly right and rigorous. We thank you for the suggestions. We have replaced the word "tellurate" with "tellurite" in the revised manuscript.

Question 2. In the main text, there are no descriptions for abbreviations for PiG and PiS. In addition, information for YAG:Ce-PiG and -PiS is insufficient. What is a component of a matrix glass of the former? How did the authors obtain the latter? In the caption of Fig. 3, the abbreviation of "phosphor in silicon" is written as PiS, but does that mean "phosphor in silicone"?

Author reply: Thanks for pointed out these problems, and we have supplemented and corrected the detailed information including abbreviations, preparation process, and component for YAG:Ce-PiG and YAG:Ce-PiS in our revised manuscript. We performed scanning electron microscopy (SEM) and energy-dispersive X-ray spectroscopy (EDS) patterns to characterize the commercial YAG:Ce-PiG, as shown in Fig. R4 and Table R2 (Supplementary Fig. 15 and Table 3 in the revised version). The sample demonstrates the glass matrix is a multi-component silicate glass.

Fig. R4 | SEM image and corresponding EDS mapping profiles of the commercial YAG:Ce-PiG (scale bar, 80 μ m).

Table R2. EDS analysis of the commercial YAG-PiG.

Element	wt%	at%
Y	14.42	3.79
Al	5.33	4.61
Si	29.7	24.7
Na	7.26	7.38
K	1.51	0.9
Ca	2.2	1.28
Ce	0.34	0.06
O	39.23	57.28

Question 3. Regarding Fig. 3d and 3e, the authors mentioned that the saturation threshold firstly increased and then decreased with increasing the concentration, possibly due to an increase of porosity resulting in a reduction of thermal conductivities and mechanical strengths. The authors should show evidence for them.

Author reply: Yes, we should discuss more details on this. As shown in the Fig. R5 (Supplementary Fig. 14 in the revised version), the change curve of sample temperature with laser power density is almost the same as that of sample saturation threshold, which indicates the higher the concentration of YAG:Ce phosphor, the more blue lasers may be down-converted in the same radiation region, generating more heat and accelerating the luminous saturation of the sample. The previous report also gave a similar phenomenon (*Ref.: Laser Photonics Rev. 16, 2200553 (2022).*). We thank the reviewers for suggesting to perform sample temperature versus variation of laser power density measurements to ensure the accuracy of the manuscript. We have made the corresponding corrections and citations in the revised manuscript.

Fig. R5 | a Curves of sample temperature versus variation of laser power density. **b** Thermal infrared images of the commercial YAG:Ce-PiG and the YAG:Ce-PGC 10 wt% samples at changed laser power density.

Question 4. In abstract, they mentioned “we develop a rapid synthetic route to PGC within ~ 10sec” and in maintext, “to achieve a dense uniform dispersion of YAG:Ce phosphor particles in tellurate glass within 5 s”. Which is insisted, 10 or 5 sec?

Author reply: In our manuscript, the temperature-time profile of rapid synthesis process for YAG:Ce-PGC samples is shown in Fig. 1a, which is divided into two stages. The first stage is to disperse YAG:Ce phosphor particles into the tellurite glass melt, which takes about 5 seconds, and the second stage is cooling and forming YAG:Ce-PGC sample, which takes about 5 seconds. Therefore, the whole process is about 10 seconds. We agree with you, and we have described it consistently in the revised manuscript.

Question 5. The temperature rise under blue laser excitation should be monitored.

Author reply: Thank you for your suggestions. We have supplemented the curves of sample temperature and corresponding thermal imaging versus variation in blue laser power density, as shown in Fig. R5 (Supplementary Fig. 14 in the revised version).

Question 6. It is understandable that the choice of tellurite glasses among various glass systems (silicate, borate, phosphate etc.) is good, as seen in Supplementary Fig.1b. But there was not provided information on why the 75TeO₂-15ZnO-5Na₂O composition was chosen.

Author reply: Yes, tellurite glasses are used in this study, and we give the details here on the choice of the composition we used, and the detailed description has been also added in the revised manuscript. For phosphor-glass composites by rapid synthetic protocol, the index of thermal stability ΔT ($\Delta T = T_x - T_g$, the larger the ΔT , the stronger the resistance to crystallization of glass) and melting temperature and viscosity in the tellurite glass system are key factors to improve the dispersivity of phosphor and reduce thermal erosion (also given in question 2 of reviewer #1). The melting temperature of glass is related to the polarizability of its constituent cations. The higher the cation polarizability, the lower the melting temperature of the glass (*Ref: Proc. R. Soc. Lond. A* **217**, 203-221 (1953)). Cations in the outer layer that contain non-inert electron pairs (such as Pb²⁺, Bi³⁺, Te³⁺, etc.) and an 18-electron configuration (such as Zn²⁺) have higher polarizability (*Ref: Prog. Mater Sci.* **57**, 1426-1491 (2012)). In addition, the introduction of ZnO is beneficial to neutralize the covalency of Te-O bond and improve the glass formation ability (*Ref.: J. Non-Cryst. Solids* **151**, 134-142 (2021)). The addition of Na₂O can further reduce the melting temperature,

drive the triangular cone $[\text{TeO}_4]$ in the glass network to $[\text{TeO}_3]$, and improve the thermal stability ΔT of the glass (**Ref.:** *J. Alloy. Compd.* 854, 157072 (2021)). Therefore, we studied the TeO_2 -ZnO- Na_2O glass system, and the composition of 75 TeO_2 -15 ZnO-10 Na_2O was screened and selected (Fig R6 and Table R3).

Fig. R6 | a-b DSC curve of TeO_2 -ZnO- Na_2O (T-Z-N) glasses.

Table R3 Composition and thermal properties of TeO_2 -ZnO- Na_2O glasses.

Element	T_g (°C)	T_x (°C)	ΔT (°C)
75 TeO_2 -20ZnO-5 Na_2O	308	349	41
75TeO_2-15ZnO-10Na_2O	279	347	69
75 TeO_2 -10ZnO-15 Na_2O	269	330	61
70 TeO_2 -25ZnO-5 Na_2O	315	365	50
70 TeO_2 -20ZnO-10 Na_2O	291	344	53
70 TeO_2 -15ZnO-15 Na_2O	271	332	61
70 TeO_2 -10ZnO-20 Na_2O	251	306	55

Question 7. Page 4, line 49. It is seen to have a simple mistyping of Fig.1g in “YAG:Ce crystals and ... (Fig.1f).”

Author reply: Thanks! The mistyping has been revised.

Question 8. Page 5, line 2-3, “.. back into the high-temperature furnace to continue heating and ..”. Please identify the temperature used here.

Author reply: Thank you for your suggestion. We have made it clear that the temperature is 650 °C in the revised manuscript.

Question 9. Page 5, line 10-12. “At 650 °C, the small wetting angle ..., preventing the atomic-scale contact and sintering of YAG:Ce particles in the melt.” Why can the high wettability (small contact angle 43.5°) prevent the atomic-scale contact? The wettability of

YAG:Ce with the tellurite glass surely gave higher dispersion of YAG:Ce particles in the tellurite glasses (Their description in Page 11, line 19-20, “the better wettability (the smaller the theta) between the particles and the molten glass in this synthesis method, the higher the interfacial energy barrier that prevent the particles from contacting each other” is correct), but that does always not mean less reactivity for YAG:Ce particles and the molten glass. The interfacial energy (barrier) will not be the barrier energy of the reaction between them.

Author reply: We addressed the above comments from the following aspects.

1. Interfacial energy barrier preventing particle atomic-scale contact and sintering: at a high temperature, YAG:Ce particles may sinter together if they are in contact with each other. The tellurite glass melt-YAG:Ce interface will be replaced by a YAG:Ce surface; the substitution between interfaces begins at the atomic scale.
2. Yes, we agree that the interfacial energy (barrier) will not be the barrier energy of the reaction between YAG:Ce particles and the molten glass. The reactivity problem is contributed to by our rapid synthesis route. First, the contact time between the particles and the tellurite glass melt is only 10 seconds (Fig. 1a). Second, our synthesis temperature is relatively low, 650°C, which slows down the interface reaction rate.

Question 10. How were the experimental parameters of barrier energy, thermal energy and attractive van der Waals energy used in the simulation found in Fig.2c? The reviewer noticed that they mentioned “the YAG:Ce particles are treated as inert-particles.” If so, the simulation presented (Fig.2c) was not any imitation of the mixing behavior with melt-particle interaction (wettability) and van der Waals interaction working between YAG:Ce particles suggested in Fig.2a.

Author reply: Yes, it is perfectly right and rigorous that with the introduction of the interface energy barrier, thermal energy, and van der Waals energy into the fluent model, it is possible to completely restore the experimental behavior; however, the model will become very complicated, and the results will not be calculated after many attempts. Therefore, to simplify the model, we treat the crystal particles as inert-particles (*Ref: J. Mater. Res. Technol. 3, 296-302 (2014).*). We think that the YAG:Ce particles are treated as inert-particles, which does not affect the innovation and rationality of our manuscript. First, we have proved the self-stabilization model of particles in glass melt under the synergistic action of interfacial energy barrier, thermal energy, and van der Waals energy through experimental and theoretical calculations (Figure 2a in manuscript). Second, the viscosity of the glass melt is one of most important factors for rapid synthesis routes. The fluent simulation demonstrates that **the low viscosity characteristics of tellurate glass melt**, which provides sufficient conditions for the complete dispersion of YAG:Ce particles only by a simple agitation strategy. Third, we utilized the shear stress transport k-omega turbulence model and the discrete phase model in ANSYS Fluent to analyze the flow characteristics of the fluid and the motion state of the particles with consideration of gravity and collisions between YAG:Ce particles in glass melt. The simulation result is consistent with the experimental result (Supplementary Video 1). The above three points fully convinced us that our manuscript was scientific and reasonable.

Question 11. No detail on the information about YAG:Ce-PiSG (silica glass?) was provided, either. Was ref.4 referred in Supplementary Fig.11 correct?

Ref.25 showed that the silica glass in YAG:Ce-PiSG was derived from triblock co-polymer and TEOS and subsequently sintered with YAG:Ce powders by SPS technique. Thus, the silica glass was not a quartz glass (melt-quenched) but a sol-gel derived silica glass. Anyway, the manuscript is difficult to be read.

Author reply: Thank you for your suggestions, we have supplemented more detailed information about YAG:Ce-PiSG in the supplementary information. Ref. 4 referred in Supplementary Fig.11 is the same reference as Ref. 25 in the manuscript. The softening point temperature of quartz glass (melt-quenched) is about 1730 °C; the phosphor-silica glass composites will be difficult to achieve by traditional pressure-assisted sintering and co-melting methods due to severe interfacial reactions at such high temperatures. Therefore, at present, phosphor-silica glass composites are achieved only by using sol-gel derived silica glass or amorphous silica nanoparticles and are reliant on spark plasma sintering and reduction sintering technology to reduce the sintering temperature and suppress interface reactions (Ref: *Acta Mater.* 130, 289-296 (2017); *Nat. Commun.* 9, 1175 (2018)).

Question 12. The authors explained, for example in page 8 line 26-27, “the rapid synthesis strategy of YAG:Ce encapsulated in dense tellurate glass can not only achieve phosphor particle integrity, ..”, while their observation was “the saturation threshold of the YAG:Ce-PGC sample first increased and then decreased, possibly due to the increased porosity in the high-doping YAG:Ce-PGC samples.” They seemed to mention that the samples were dense and porous, which was difficult to be understood: The latter observation was true for the 10wt% doping as well as for 20wt% doping, as seen in Fig.3d. That means the 10wt% doped PGC sample also had higher porosity and degraded thermal conductivity, if the authors’ claim was correct. It is contradiction and confusing.

Author reply: We mentioned that "the saturation threshold of the YAG:Ce-PGC sample first increased and then decreased, possibly due to the increased porosity in the high-doping YAG:Ce-PGC samples," which means that the pores are introduced with YAG:Ce phosphor rather than the glass matrix itself being porous. Therefore, PGC samples with high phosphor doping concentrations may have higher porosity. As can be seen from Fig. R7, the pores appear mainly in the proximity of phosphor particles for all four methods. (Ref.: *J. Korean Ceram. Soc.* 56, 71-76 (2019); *Opt. Mater. Express* 7, 4304-4315 (2017); *J. Lumin.* 214, 116531 (2019).). Therefore, our descriptions are not contradictory. However, the saturation threshold increases and then decreases, mainly due to higher YAG:Ce concentration allows down-converting more blue light will produce more heat in the same radiated area, accelerating luminescence saturation of the sample (Fig. R5, Supplementary Fig. 14 in the revised version) (also answered in question 3)). We thank the reviewers for suggesting to perform sample temperature versus variation of laser power density measurements to ensure the accuracy of our representations. We have made corresponding corrections and citations in the revised manuscript. It will be clear in this revised manuscript. Thanks for the suggestions and comments.

[redacted]

Fig. R7 | SEM images of YAG:Ce-PGC prepared by our rapid synthetic route, gas pressure sintering (GPS), spark plasma sintering (SPS), and Co-melting process (CMP). Note that SEM images of PiGs using SPS, GPS, and CMP were quoted from the literatures (**Ref.:** *J. Korean Ceram. Soc.* 56, 71-76 (2019); *Opt. Mater. Express* 7, 4304-4315 (2017); *J. Lumin.* 214, 116531 (2019)). Red circle represents where the pores are located.

Question 13. Please more clearly identify the optimized concentration of YAG:Ce powders for the fabrication of YAG:Ce-PGCs. It must be 10wt%.

Author reply: First, it can be seen from Figure 3b that the YAG:Ce-PGC 10 wt% sample has the highest internal/external quantum efficiency (98.4 / 85.4%). Second, YAG:Ce-PGC 10 wt% sample shows the maximum values of luminous flux and the laser saturation threshold are 1227 lm, 8.5 W mm⁻¹, respectively. Third, YAG:Ce-PGC 10 wt% sample has a decent transmittance (about 40%). Therefore, we believe that YAG:Ce-PGC 10 wt% sample is an optimal choice.

Question 14. They mentioned “the corresponding CRI and CCT decreased with increasing doping concentration, mainly due to the increase in yellow components (Supplementary Fig.13).” To see it more closely, the 5 and 10wt% concentrations gave almost the same properties but the 15wt% decreased the CRI and CCT in comparison with the lower concentrations. These behavior was almost identical for the 20wt% concentration, as well. Careful description would be required.

Author reply: Yes, careful description is required and we have added more explanations in the revised manuscript. The sudden drop of CRI and CCT is attributed to the synergy of two factors. First, as described in our original manuscript, the increase in YAG:Ce content led to a decrease in CCT and CRI. Second, PGC samples with higher phosphor doping concentrations may have higher porosity, which will affect the results of CRI and CCT due to increasing light scattering centers. As shown in Fig. R8 (Supplementary Fig. 17 in the revised version), for the YAG:Ce-PGC 15 wt% sample, there is an obvious increase in porosity compared to the YAG:Ce-PGC 10 wt% sample, resulting in a serious scattering

phenomenon. According to your suggestion, we have also corrected the corresponding explanations in the revised manuscript.

Fig. R8 | a-d SEM image and corresponding EDS mapping profiles of the YAG:Ce-PGC with 5-20 wt% YAG:Ce phosphor (scale bar, 80 μm).

Question 15. In Fig.3d, a part of data was hidden by the legend of the figure. Please revise it.

Author reply: Thank! We have corrected it in the revised manuscript.

Finally, we thank you for the constructive and valuable comments again.

Reviewer #3 (Remarks to the Author):

Comments. I co-reviewed this manuscript with one of the reviewers who provided the listed reports. This is part of the Nature Communications initiative to facilitate training in peer review and to provide appropriate recognition for Early Career Researchers who co-review manuscripts.

Response. We thank this Reviewer for co-reviewed our manuscript with one of the reviewers. Thanks for providing constructive and valuable comments. According to your suggestions, we have made relevant revision. The revised sentences were highlighted with blue font color in the revised manuscript.

Reviewer #4 (Remarks to the Author):

Comments. The manuscript „Rapid synthesis of phosphor-glass composites in seconds” presents an interesting experimental investigation of YAG:Ce particles uniformly embedded in a glass matrix. The authors have carried out extensive and careful experimental work and meaningful analytical studies of the samples, taking into account the relevant literature and discussing the results with theoretical models. The extensive appendix supports the understanding and interpretation of the results. The work has practical relevance and may help to solve the problem of degradation of phosphors at high irradiation energy. The manuscript is recommended for publication in Nature Communications with the following minor corrections:

Response. We are very grateful to the Reviewer for the positive comments. According to your suggestions, we have also made relevant revision. The revised sentences were highlighted with blue font color in the revised manuscript.

Question 1. Pg. 4 : the numerical value of the refractive index tellurate glass (1.82) and YAG:Ce crystal (1.97) was exchanged

Author reply: This is right. Thanks for spotting this typo, which has been fixed.

Question 2. Fig. 2a: the visibility of the wetting angle in the graphic is poor.

Author reply: Thanks for your valuable comments. We have optimized the wetting angle graphic in Fig. 2a in the revised manuscript.

Question 3. Fig. 2c: use r/s instead of s

Author reply: The fluent simulation in Fig. 2c is performed under 5 r/s, and other examples are given in Supplementary Fig. 10. Several figures in Fig. 2c demonstrate different stages at 0, 1, 3 and 5 s, so that we need not change it here. The description in the figure caption herein is not clear, and we have modified the figure captions of Fig. 2c.

Question 4. Pg. 5: “to continue heating”.. at 650°C?

Author reply: Yes, you are right. Thank you for your suggestion. We have made it clear that the temperature is 650 °C in the revised manuscript.

Question 5. Fig. 3f: delete “, and YAG:Ce phosphor in glass,” in the caption

Author reply: Thank you! We have corrected it in the revised manuscript.

Finally, we thank you for the constructive and valuable comments again.

REVIEWERS' COMMENTS

Reviewer #1 (Remarks to the Author):

The authors have responded to the questions and points raised by the reviewers. I recommend publication.

I hope that the exchange with the reviewers can be published as well. If not I encourage the authors to add the extra figures and tables prepared for the reviewers into the SI section of the manuscript.

Best regards,
Mathieu Allix

Reviewer #2 (Remarks to the Author):

The authors have revised the manuscript along with the reviewers' comments. However, incompleteness's been found, which should be corrected. To make it much better, the following points are further to be revised before recommending it for publication in Nat.Commun.

(1) The revised manuscript still has the term of "tellurate". Unless it is corrected, the paper will become confusing. Please carefully check the figures prepared. Fig.1d (Tellurate glass), Fig.2b (Tellurate glass melt), Supplementary Figs.3 and 7.

(2) The authors mentioned in the revised manuscript "To better understand the physical process, a theoretical analysis was performed." in the left column on page 3 and subsequently "We utilized the shear stress transport k-omega turbulence model and the discrete phase model in ANSYS Fluent to analyze the flow characteristics of the fluid and the motion state of the particles." in the right column on page 3. BUT, as the authors agreed in the answer to Question 10 (Reviewer#2), the fluid modeling did NOT include any analysis of wettability between the melt and YAG particle. So, ".. in ANSYS Fluent to analyze the flow characteristics of the fluid" should be correctly revised to ".. in ANSYS Fluent to simulate the flow characteristics of the fluid." The same sentence is found in Method - Fluent (Fluid?) simulation model description.

(3) For Supplementary Fig.17, the appropriate figure caption should be given for (a), (b), (c), and (d)., for example, (a) 5wt%, (b) 10wt%, (c) 15wt%, and (d) 20wt%. Otherwise, readers cannot correctly understand what (b) and (c) are.

(4) The term "silicone" is better to be expressed with "silicone resin", (I) in the caption of Fig.3., in "YAG:Ce phosphor in silicone (YAG:Ce-PiS)" (II) in Performance characterizations of YAG:Ce-PGS and (III) in Methods

(5) In Method - Fluent simulation model description, "Supplementary Fig. 9a" should be "Supplementary Fig.10a".

(6) In Method – YAG:Ce particles self-stabilization model, "As shown in Fig.3a, at 923 K,.. " should be "As shown in Fig.2a, at 923 K,.. "

(7) Supplementary Table 2, "LuAG powder" and "GdAG powder" should be "LuAG:Ce powder" and "GdAG:Ce powder", respectively.

Reviewer #3 (Remarks to the Author):

Reviewer #1 (Remarks to the Author)

Comments. The authors have responded to the questions and points raised by the reviewers. I recommend publication.

I hope that the exchange with the reviewers can be published as well. If not I encourage the authors to add the extra figures and tables prepared for the reviewers into the SI section of the manuscript.

Best regards,

Mathieu Allix

Response. Thank you very much for the positive recommendation. Of course, we are pleased to share our exchange, and we will upload our review comments publicly.

Reviewer #2 (Remarks to the Author):

Comments. The authors have revised the manuscript along with the reviewers' comments. However, incompleteness's been found, which should be corrected. To make it much better, the following points are further to be revised before recommending it for publication in Nat.Commun.

Response. We thank the Reviewer for his/her comments which helped us to improve our study. According to your suggestions, we have made relevant revisions. The revised sentences were highlighted with **blue font color** in the revised manuscript.

Question 1. The revised manuscript still has the term of “tellurate” . Unless it is corrected, the paper will become confusing. Please carefully check the figures prepared. Fig.1d (Tellurate glass), Fig.2b (Tellurate glass melt), Supplementary Figs.3 and 7.

Author reply: Thank you! We have carefully checked and corrected this problem for our revised manuscript.

Question 2. The authors mentioned in the revised manuscript “To better understand the physical process, a theoretical analysis was performed.” in the left column on page 3 and subsequently “We utilized the shear stress transport k-omega turbulence model and the discrete phase model in ANSYS Fluent to analyze the flow characteristics of the fluid and the motion state of the particles.” in the right column on page 3. BUT, as the authors agreed in the answer to Question 10 (Reviewer#2), the fluid modeling did NOT include any analysis of wettability between the melt and YAG particle. So, “.. in ANSYS Fluent to analyze the flow characteristics of the fluid” should be correctly revised to “.. in ANSYS Fluent to simulate the flow characteristics of the fluid.” The same sentence is found in Method - Fluent (Fluid?) simulation model description.

Author reply: Yes, this is perfectly right and rigorous. We have corrected it in the revised manuscript according to your suggestion.

Question 3. For Supplementary Fig.17, the appropriate figure caption should be given for (a), (b), (c), and (d)., for example, (a) 5wt%, (b) 10wt%, (c) 15wt%, and (d) 20wt%. Otherwise, readers cannot correctly understand what (b) and (c) are.

Author reply: Thank you for your suggestions. We have corrected it in the revised manuscript.

Question 4. The term “silicone” is better to be expressed with “silicone resin” , (I) in the caption of Fig.3., in “YAG:Ce phosphor in silicone (YAG:Ce-PiS)” (II) in Performance characterizations of YAG:Ce-PGS and (III) in Methods

Author reply: Thank you! We have corrected it in the revised manuscript.

Question 5. In Method - Fluent simulation model description, “Supplementary Fig. 9a” should be “Supplementary Fig.10a” .

Author reply: Thanks! The mistyping has been revised.

Question 6. In Method – YAG:Ce particles self-stabilization model, “As shown in Fig.3a, at 923 K,.. ” should be “As shown in Fig.2a, at 923 K,.. ”

Author reply: Thanks for spotting this typo, which has been fixed.

Question 7. Supplementary Table 2, “LuAG powder” and “GdAG powder” should be “LuAG:Ce powder” and “GdAG:Ce powder” , respectively.

Author reply: Thank you! We have corrected it in the revised manuscript.

Finally, we thank you for the constructive and valuable comments again.

Reviewer #3 (Remarks to the Author):

Comments. I co-reviewed this manuscript with one of the reviewers who provided the listed reports. This is part of the Nature Communications initiative to facilitate training in peer review and to provide appropriate recognition for Early Career Researchers who co-review manuscripts.

Response. We thank this Reviewer for co-reviewed our manuscript with one of the reviewers. Thanks for providing constructive and valuable comments. According to your suggestions, we have made relevant revision. The revised sentences were highlighted with **blue font color** in the revised manuscript.